# Bone Modeling after Orthodontic Extrusion: A Histomorphometric Pilot Study

**DOI:** 10.3390/jcm11247329

**Published:** 2022-12-09

**Authors:** Marco Montevecchi, Gianluca Marucci, Barbara Pignataro, Gabriela Piana, Giulio Alessandri-Bonetti, Vittorio Checchi

**Affiliations:** 1Department of Biomedical and Neuromotor Sciences—DIBINEM, University of Bologna, Via San Vitale 59, 40100 Bologna, Italy; 2Fondazione IRCCS Istituto Neurologico Carlo Besta, 20133 Milan, Italy; 3Private Practice, Casalecchio di Reno, 40033 Bologna, Italy; 4Department of Surgery, Medicine, Dentistry and Morphological Sciences, Unit of Dentistry and Oral-Maxillo-Facial Surgery, University of Modena and Reggio Emilia, Via del Pozzo 71, 41100 Modena, Italy

**Keywords:** bone modeling, histology, orthodontic extrusion, orthodontic extraction, tooth retention

## Abstract

During osteogenesis and bone modeling, high vascularity and osteoblastic/osteoclastic cell activity have been detected. A decrease in this activity is a sign of complete bone formation and maturation. Alveolar bone maturation seems to occur within weeks and months; however, the precise timing of the alveolar bone modeling is still unknown. The aim of this clinical pilot study was to investigate the bone modeling of neo-apposed tissue during orthodontic extrusive movements, through a histomorphometric analysis of human biopsies. This study was conducted on third mandibular molars sockets, and all teeth were extracted after orthodontic extrusion between 2010 and 2014. After different stabilization timings, extractions were performed, and a specimen of neo-deposed bone was harvested from each socket for the histomorphometric analysis. Histological parameters were evaluated to identify bone quantity and quality. This study included 12 teeth extracted from 9 patients. All specimens were composed of bone tissue. Bone samples taken after 1 and 1.5 months of stabilization presented remarkable percentages of woven bone, while after 2 months, a relevant decrease was observed. Histomorphometric analysis suggested that after orthodontic extrusion, a period of stabilization of 2 months allows the neo-deposed bone to mature.

## 1. Introduction

Tooth movement induced by the application of orthodontic forces is characterized by the remodeling of periodontal tissues [1]. Depending on the direction of the applied force, compression zones and tension zones are created on all the tissues involved, which in turn respond according to specific adaptation mechanisms [2,3]. Alveolar bone response to orthodontic forces is expressed with tissue resorption by osteoclasts in compressed sites, and with tissue neo-deposition by osteoblasts in tension sites [4].

The first physiologic step in the bone neo-deposition process consists of the formation of woven bone, which is further resorbed and immediately replaced by lamellar bone [5]. These two bone matrixes differ in their predominant collagen fibers arrangement: in woven bone, collagen fibrils are randomly oriented, while in lamellar bone, they are clustered in parallel arrays and alternated in orthogonal patterns [6]. During osteogenesis and bone remodeling, high vascularity and contemporary activity of osteoblastic and osteoclastic cells have been detected [7]. A decrease in vascularity and osteoclastic activity are signs of complete bone formation and maturation [7] and, therefore, of the achievement of tissue stability.

At the end of an orthodontic movement, the whole periodontal complex undergoes a gradual maturation process that results in a re-establishment of the initial physiological architecture [2]. Periodontal ligament reorganization seems to occur in 3–4 months, corresponding to the clinical recovery of physiological tooth mobility [8]. However, histological observations have shown that gingival fibers need about 6 months to reorganize themselves [9]. Concerning alveolar bone maturation timing, an extensive period ranging between weeks and months has been reported [5]. Consequently, the precise maturation timing of the alveolar bone is still unknown.

The degree of bone maturation after orthodontic forces can have clinical implications such as the influence on the final tooth-position stability. The identification of the exact timing of tooth retention after active orthodontic therapy can be relevant in order to avoid a potential relapse. Furthermore, in the last few years, several authors have suggested the use of orthodontics to recreate an ideal bone condition for implant positioning [10,11]. Understanding the maturation timing of the bone induced by this approach is crucial for a stable and successful result.

In light of these considerations, deepening the knowledge on this topic can certainly be relevant.

The “orthodontic extraction” (OE) technique was introduced in 1996 for the management of impacted third molars in a close anatomical relationship with the mandibular canal [12]. As the roots of these teeth are pulled away from the mandibular canal by means of orthodontic movement, the risk of neurological damage is greatly reduced, making subsequent extraction easier, quicker, and safer [13]. The orthodontic extrusive movement produces tensional forces on the periodontal fibers of the third molars, thereby resulting in new bone apposition along the path of tooth eruption and periodontal healing distal to the second molar [14].

The aim of this retrospective clinical pilot study was to investigate the maturation timing of neo-apposed bone during orthodontic extrusive movements, through a histomorphometric analysis of human bone biopsies.

The secondary outcome was to compare the microstructural features of the newly formed bone, induced by orthodontic extrusion, using the data present in the literature on surgical bone-augmentation procedures.

## 2. Materials and Methods

The study sample was made of 12 consecutive impacted mandibular third molars, orthodontically extruded, and subsequently extracted, as stated by the OE protocol, at the University of Bologna (Italy) between 2010 and 2014. For each of these cases, a bone core had been harvested from the alveolar socket immediately after the tooth extraction. Specimens had been fixed in formalin for subsequent histomorphometric analyses.

In order to be included in the study, patients had to be older than 18 years and without any systemic conditions, nor receiving any pharmacologic therapies that could interfere with the bone-modeling processes. The extracted third molar had to be initially symptomatic, although not displaying any sign of endodontic pathology. All treated molars were deemed to be extracted due to pericoronitis, or periodontal damage to the adjacent mesial tooth. Ethics committee approval was obtained (CE code: 17048, International Bologna–Imola ethical committee) and subjects signed an informed consent form.

### 2.1. Clinical Procedures

All patients had undergone OE of the impacted teeth using a previously described standardized protocol [12,13,14]. This approach finds its specific indication for those third molars in tight contact with the mandibular canal. This condition was present in all third molars enrolled in the present study and confirmed by a CT scan exam performed at the initial visit. In all cases, a second CT scan was performed after the OE to confirm the desired distance (at least 1 mm) between the tooth and the mandibular canal. Only at this point was the extraction performed, with no risks of damaging the alveolar nerve [12,13,14]. Different orthodontic techniques were applied: dental anchorage with nickel–titanium arch wire, dental anchorage with sectional stainless-steel wire (cantilever), and skeletal anchorage with mini-screws (Figure 1).

In the case of tooth interference with the occlusion, this issue was solved through an occlusal adjustment. The OE was justified by the close proximity between the third molar anatomy and the mandibular canal. All treated molars were in partial or total mucosal inclusion and were multirooted. However, the clinical decision to use the OE technique was based primarily on the chance to avoid nerve damage risks during a surgical extraction, and to obtain excellent periodontal healing distal to the second molar [14]. After a period of about 3–6 months, once the roots have moved away from the canal, the orthodontic appliance was passivated and left in place for a variable retention phase: from one to seven months. After the stabilization period, extraction was performed by the same experienced periodontist and a specimen of neo-deposed bone was harvested from the socket with a 2 mm diameter trephine bur. Particular care for the alveolar bone socket was maintained during the extractive procedure, excessive forces or useless osteotomies were avoided, and bone substitutes or membranes were not used.

In order to collect the newly formed bone, the trephine bur orientation was guided by the main axis of the tooth movement and induced by the extrusive force. The location and axis planning of the biopsy were defined through the final CT scan. Bone samples of at least 3 mm in length were collected and immediately stained on the most apical portion to assure a correct tissue orientation. Extreme care was applied so as not to interfere with the mandibular canal.

### 2.2. Histologic Procedures

Bone biopsies were fixed in 10% buffered neutral formalin (Bio-Optica, Milan, Italy) for 24 h. The formalin-fixed tissue specimens were water washed for 30 min before being placed in the decalcifying agent (EDTA) (MicroDec, Diapath, Martinengo, Italy) at room temperature. The solutions were changed periodically until decalcification was achieved. After the decalcification procedure was completed, the tissues were washed in distilled water (Tank of distilled water, Bio-Optica, Milan, Italy) in order to remove the decalcifying agent.

Successively, tissues were dehydrated and embedded in paraffin (Bio Plast Waxes, Bio-Optica, Milan, Italy). Serial 4 μm-thick longitudinal sections were cut and stained with hematoxylin and eosin (Diapath, Martinengo, Italy) (H & E) for microscopic examination.

For each sample, at least three sections, one every 200 μm, were evaluated. Overall, thirty-six H & E-stained sections were microscopically analyzed. The Aperio ScanScope digital slide-scanner (Aperio, Vista, CA, USA) was used to scan the entire stained slide at 20× magnification, in order to create a single high-resolution digital image and to perform a histometric evaluation.

The histological parameters evaluated to identify the bone quantity were: the bone volume (defined by the percentage of bone tissue (both mineralized and osteoid) on the total volume of the examined tissue: Bone Volume ÷ Tissue Volume × 100), and the connective tissue volume (defined by the percentage of connective tissue on the total volume of the examined tissue: Connective Tissue ÷ Tissue Volume × 100) [15]. 

The histological parameters evaluated to identify bone quality were: the mineralized bone volume (defined by the percentage of mineralized tissue on the bone volume: Mineralized Tissue ÷ Bone Volume × 100); osteoid volume (Osteoid Tissue ÷ Bone Volume × 100); woven bone volume (Bone Woven ÷ Bone Volume × 100); lamellar bone volume (Bone Lamellar ÷ Bone Volume × 100); trabecular bone volume (Bone Trabecular ÷ Bone Volume × 100); bone marrow volume (Bone Marrow ÷ Bone Volume × 100), and osteoblasts/osteoclasts ratio [16]. The vascularization was assessed by evaluating, for each slide, the number of blood vessels × mm^2^, and averaging the value per patient.

### 2.3. Statistical Analyses

The Shapiro–Wilks test was used to compare the histological parameters evaluated with the Gaussian distribution and their description was made by the means of the median and interquartile range. Assuming the existence of a trend between the examined periods of stability and bone parameters, the significance of this trend was evaluated with the Jonckheere–Terpstra test. The significance level ‘alpha’ has been fixed to 0.05.

## 3. Results

This study included nine patients (all females; mean age: 35 ± 13 years; one smoker), from which, a total of 12 alveolar sockets were analyzed (Table 1).

In some specimens, it was not possible to determine the osteoblast/osteoclast ratio because of the presence of cell necrosis, artifacts, or the unique osteoblast or osteoclast cell typology. In only three samples, was a prevalence of osteoblastic cells observed. Table 2 reports the interquartile ranges of the evaluated bone parameters, denoting the variability of the same parameters.

Concerning histologic tissue quality, all the specimens examined were composed of bone tissue in a percentage ranging from 80% to 100% (in three cases, it was present for about 10–20% of the connective tissue). Evaluating the stabilization timings for 1, 1.5, 2, 3, 6, and 7 months in association with bone variables, a significant trend resulted only for the woven bone volumes (*p* = 0.049). Bone samples taken after 1 and 1.5 months of stabilization (Figure 2 and Figure 3) presented a high percentage of woven bone (30% and 20%, respectively).

While at 2 months, Figure 4 shows 5% woven bone and increased vascularization.

It was also observed that in the specimens, in which the osteoblast prevailed over the osteoclast, the median of the period of stability was 2 months.

## 4. Discussion

This clinical case series study analyzed a cohort of nine subjects (12 teeth) treated with OE [12,13,14], aiming to define the maturation timing of neo-apposed bone during orthodontic movements. Using the mean from the histomorphometric analysis, the neo-deposed bone was studied at different stabilization timings of 1, 1.5, 2, 3, and 7 months after OE. The slight number of specimens and the small number of bone biopsies represent an undeniable limitation of the present study. Another limiting aspect is the broad age range of the enrolled patients and the fact that only females were part of the study sample. Finally, the heterogeneity of the orthodontic appliances used for OE, and the retrospective study design, are further relevant aspects that make future and more structured investigations strongly advisable. 

Moreover, very few studies have been published on this topic, and in particular on human samples. Therefore, the present findings must be considered as suggestions for further research and insights into the modeling of orthodontically induced bone.

In orthodontics, a retention period seems to be necessary in order to allow the correct maturation of the newly formed bone. Studies on animal models by Van Venrooy & Yukna and Berglundh et al. stated that 21 days and 8 weeks, respectively, were the ideal periods for bone modeling after tooth extrusion [17,18]. A few further case reports in humans tried to show maturation periods of 8 weeks and 2 and 3 months, respectively [19,20,21]. More recently, a narrative review about forced orthodontic eruption concluded that further studies are necessary to elucidate the long-term stability of orthodontically extruded teeth and the supporting bone that follows them [22].

Various biological models, similar to orthodontic extrusions, such as alveolar distraction osteogenesis and bone grafting in extraction sockets, could be compared concerning bone maturation processes. Histomorphometric analysis showed mature lamellar bone 70 days after alveolar distraction procedures [23]. In tooth-extraction sites, the socket is remodeled into the marrow and lamellar bone. A study on animal models revealed a percentage of 75% bone marrow after 90 days, increasing to 85% at 180 days [24]. A histological pilot study on humans instead showed lower percentages (54%) of mature bone after 6 months [25]. A more recent paper showed that 6 months after extraction and socket preservation, there was 17% lamellar bone and 16% woven bone, whereas there was 35% lamellar bone and 10% woven bone after post extractive guided bone regeneration [26].

In the present retrospective study, woven bone was observed in high percentages after 1 and 1.5 months, with a significant decrease at two months. The low quantity of woven bone after 2 months of stabilization could indicate an almost-reached bone maturation and stability after orthodontically induced bone remodeling. To support the hypothesis that after this period, the osteogenesis could definitely be concluded is the observation that for the specimens where osteoblasts prevailed over osteoclasts, the median stability period was 2 months. Therefore, it can be speculated that 2 months could represent the correct period of stabilization required after orthodontic forces, in order to reach complete osteogenesis.

Differently from the nonfunctionally loaded bone, neo-deposed bone after orthodontic tooth movement seems to have a significantly higher woven bone volume (10%) and a lower bone-marrow volume (0%). The greater presence of woven-bone volume might be explained by the activation of bone neo-deposition in response to the forces transmitted by the stretching of the periodontal ligament, and by the need to acquire plasticity rather than hardness in order to withstand tensile stress without fracturing. The absence of bone marrow represents a bone density increase index. In animal models, it has been shown that in tension sites, alveolar bone becomes denser through a decrease in trabecular separation and an increase in bone volume fraction [27].

A limitation of the present histological evaluation is represented by its shortcomings in the employment of emerging immunohistochemical osteogenic markers, which are becoming increasingly available.

Micro-computed tomography (µCT) is currently used to evaluate morphometric bone characteristics as an alternative to conventional histological analysis [28]. Using µCT allows a more representative analysis of the entire sample, although it should be highlighted that histology remains the most indicated method to evaluate proteins, cells, and composition [28].

In a recent study on an animal model, after tooth extraction, characterization of the alveolar bone healing was performed by µCT [29]. After 28 days of healing, µCT showed the newly formed bone had favorable morphometric characteristics of quality and quantity, whereby the bone volume and trabecular thickness parameters progressively increased from 7 days of healing to 28. Consequently, a gradual decrease in trabecular separation, trabecular space, and total bone porosity was observed. [29].

Similar trends were reported by another animal model study, designed to investigate the process of calcification during bone healing, as measured by bone mineral density [30]. The mineral density of the healing bone, evaluated by µCT, increased with time, and the healing bone became thicker and denser between 7 and 56 weeks of the healing period [30].

The exact explanation for the maturation timing of the alveolar bone, and better knowledge of the specific characteristics of this newly deposed bone, could be significant for different clinical aspects. It could be very useful to establish an exact retention period, from which the bony architecture is biomechanically stable, in order to prevent post-treatment relapses or unwanted teeth migrations [31].

Moreover, the importance of bone quality and quantity for osseointegration of dental implants is well documented [32,33,34]. Having knowledge of the exact bone-modeling phases after orthodontic forces suspension could be very useful to indicate the ideal time to insert an implant [10,35,36].

## 5. Conclusions

Information obtained from this histomorphometric analysis suggests that, after orthodontic extrusion, a period of stabilization of 2 months can allow the neo-deposed bone to mature. The achievement of tissue maturation and stability was represented by 0–10% woven bone and the predominant osteoblastic population. Furthermore, it was observed that, in contrast with the dental movement, the neo-deposed bone obtains a typically compact histological structure, represented by 0% bone marrow. To confirm these observations, prospective studies on larger samples and defined stabilization times are strongly suggested.

## Figures and Tables

**Figure 1 jcm-11-07329-f001:**
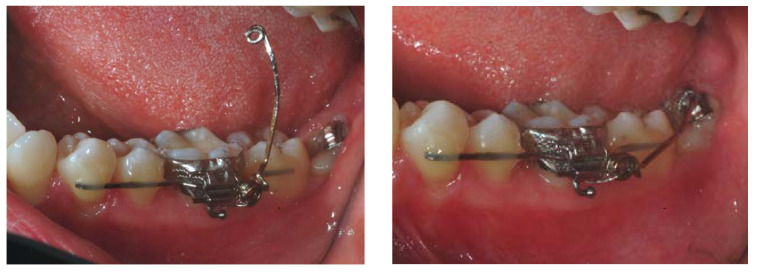
Clinical view of an example of an orthodontic appliance used for OE.

**Figure 2 jcm-11-07329-f002:**
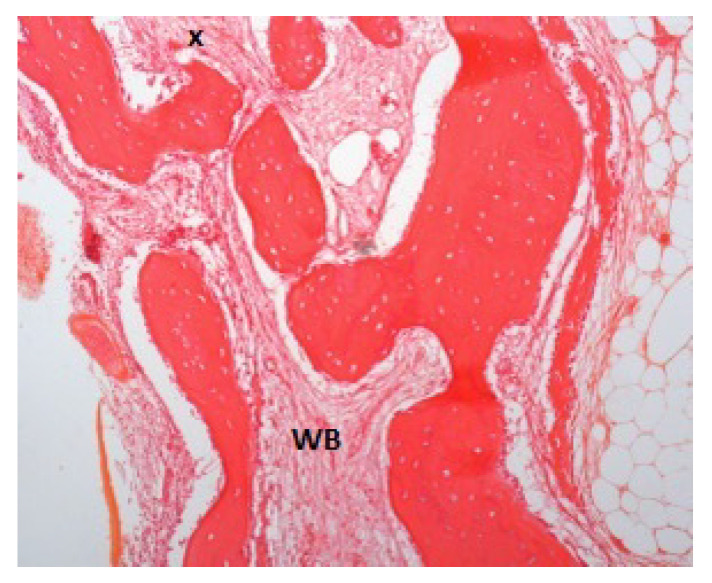
Patient ID: PD. Bone samples taken after 1 month of stabilization presented 70% woven bone (WB). The small black cross indicates the apical portion of the sample.

**Figure 3 jcm-11-07329-f003:**
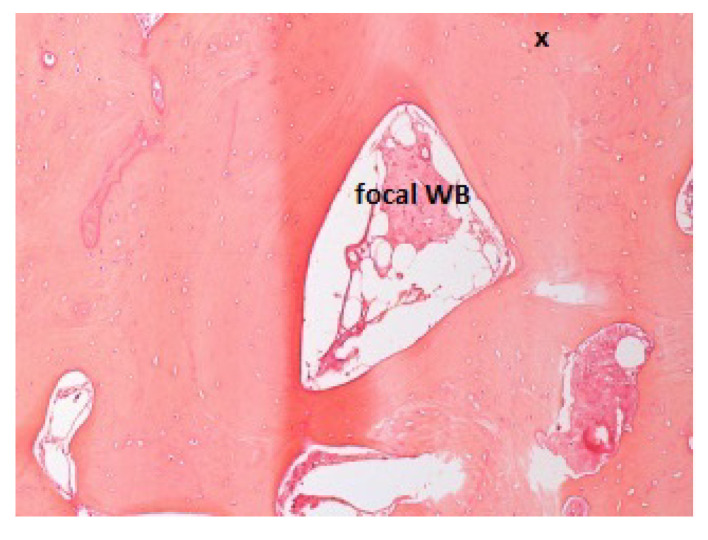
Patient ID: MM. Bone samples taken after 1.5 months of stabilization presented 20% woven bone (WB). The small black cross indicates the apical portion of the sample.

**Figure 4 jcm-11-07329-f004:**
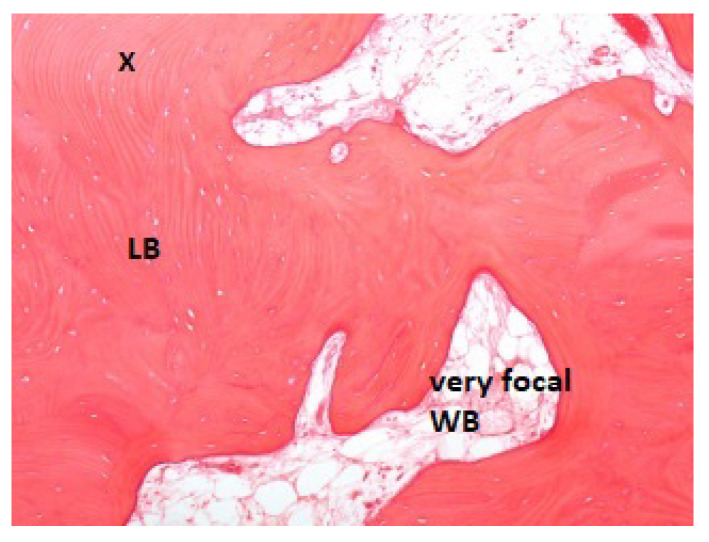
Patient ID: MA. Bone samples taken after 2 months of stabilization presented 10% woven bone (WB). LB: Lamellar bone. The small black cross indicates the apical portion of the sample.

**Table 1 jcm-11-07329-t001:** Patients-related features (all females, all ASA I).

Patient Id *n* = 9	Age	Tooth Number	Tooth Inclination	Smoke	Drugs
**A**	21	48	Horizontal	No	No
		38	Vertical		
**B**	21	48	Vertical	No	Birth-control pill
**C**	28	48	Vertical	No	Birth-control pill
**D**	30	48	Horizontal	No	Levothyroxine sodium
**E**	31	38	Horizontal	No	No
**F**	35	38	Vertical	No	No
		48	Vertical		
**G**	39	48	Vertical	Yes(5 cig/die)	No
**H**	45	38	Vertical	No	No
		48	Vertical		
**I**	62	38	Vertical	No	No

Percentages of the bone parameters examined are available as Appendix A.

**Table 2 jcm-11-07329-t002:** Median percentages of the osseous parameters, and in parentheses, the interquartile range of the bone parameters (*n* = 12).

Bone Parameters	Median Percentage (%)	Interquartile Range (%)
** *Bone tissue (%)* **	100	92.5–100
** *Connective tissue (%)* **	0	0–7.5
** *Mineralized bone (%)* **	80	70–90
** *Osteoid tissue (%)* **	10	6.25–27.5
** *Woven bone (%)* **	15	2.5–37, 5
** *Lamellar bone (%)* **	69.5	15–80
** *Trabecular bone (%)* **	0	0–0
** *Bone marrow (%)* **	0	0–0

## Data Availability

Data supporting reported results can be found at m.montevecchi@unibo.it.

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
