# Peer review of "Bone Modeling after Orthodontic Extrusion: A Histomorphometric Pilot Study"

_jcm, 2022, doi:10.3390/jcm11247329_

Round 1

Reviewer 1 Report

Investigation of the maturational timing of bone induced by tooth extrusion is of great interest for orthodontists, surgeons and prosthodontists. In that sense, this is a very useful investigation, particularly because of the lack of similar investigations on human material. However, the Materials and Methods section needs some more detailed information on the 3rd molars used for OE.

In what way is this a retrospective study? Do you collect bone specimens from every patient? What was the reason for extraction; what type of orthodontic appliance was used, and could you supply a photo. Were the molars erupted before the procedure, or not? Were they multi rooted or one rooted? For how long were they extruded – important to know since the length of extrusion could have impact on bone formation and type. What does it mean “reached the planned position”? Did the extruded tooth interfere with the occlusion?

It is mentioned that care has been taken not to interfere with the mandibular canal-this must have been extremely difficult. How did you manage to harvest 3mm bone in length – how “deep” was it? Is 3mm of bone enough to get an idea of what is happening apically to a molar? Were the “blocks” equal in size and which part of the block was used for evaluation? Could you ensure that it was always the most apical bone of the socket that was harvested? What about photos of samples with 5-7 months of stabilization? It would have been nice for comparison.

If this is a pilot study, what do the authors plan to do further?

Author Response

- Investigation of the maturational timing of bone induced by tooth extrusion is of great interest for orthodontists, surgeons and prosthodontists. In that sense, this is a very useful investigation, particularly because of the lack of similar investigations on human material. However, the Materials and Methods section needs some more detailed information on the 3rd molars used for OE.

Dear Reviewer, thank you for the comment undoubtedly aimed at making more understandable both our decision-making and the procedure with which the 3rd molars were treated. OE is a standardized protocol with precise indications. We added a more detailed description in lines 86-88, 93-98, and 104-110.

- In what way is this a retrospective study? Do you collect bone specimens from every patient? What was the reason for extraction; what type of orthodontic appliance was used, and could you supply a photo. Were the molars erupted before the procedure, or not? Were they multi rooted or one rooted? For how long were they extruded – important to know since the length of extrusion could have impact on bone formation and type. What does it mean “reached the planned position”? Did the extruded tooth interfere with the occlusion?

Thank you for these questions. No, we do not collect bone specimens from every patient, but we did so in these specific cases in order to start studying the bone maturation after OE. This happened because our research group created this safe extraction technique of impacted lower wisdom teeth in contact with the alveolar canal back in 1996 (doi: 10.14219/jada.archive.1996.0413 – Ref. #12). Over the years, we have tried to improve and make this procedure reproducible, and with this in mind we have taken the samples in question (from 2010 to 2014). Only later on we considered appropriate to publish these data and, consequently, we requested the approval of the ethics committee as a retrospective study, i.e. as a study showing data obtained in a period prior to that in which the scientific work was produced.

All treated molars were deemed to be extracted due to pericoronitis or periodontal damage to the adjacent mesial tooth. We added this sentence (Lines 86-88). The decision to approach these cases using OE technique was based on the fact that the roots of all these teeth were in contact with the alveolar canal, and that a “traditional” surgical extraction could have caused nerve damage (doi: 10.1016/j.joms.2007.06.686 – Ref. #13).

The orthodontic appliance used for OE is chosen based on the single features of each clinical case. Different orthodontic techniques can be applied: dental anchorage with nickel-titanium arch wire or with sectional stainless-steel wire (cantilever), but also skeletal anchorage through mini-screws (Lines 99-101).

We added a clinical image of an orthodontic appliance as requested (Fig. 1, page 3).

All treated molars were in partial or total mucosal inclusion and were multi-rooted. We added this explanation (Line 106-110).However, the clinical decision to use OE technique was based primarily on the chance to avoid nerve damage risks during a surgical extraction and to obtain excellent periodontal healing distal to the second molar (doi: 10.1016/j.ijom.2014.03.015 - Reference #14).

With this in mind, when we mention “reached the planned position” we mean that the roots are not in contact anymore with the alveolar canal. In case of tooth interference with the occlusion, this issue was solved through an occlusal adjustment. We modified the text as suggested and added more explanations (Lines 96-98 and 104-105).

As correctly stated, the length of extrusion could have impact on bone formation and type. Length of extrusions usually depends of the amount of orthodontic traction needed to avert the roots from the alveolar canal. For this reason, it depends and is never the same. The third molars included in this study were extruded for a period ranging from 3 to 6 months (Lines 109-110).

- It is mentioned that care has been taken not to interfere with the mandibular canal-this must have been extremely difficult. How did you manage to harvest 3mm bone in length – how “deep” was it? Is 3mm of bone enough to get an idea of what is happening apically to a molar? Were the “blocks” equal in size and which part of the block was used for evaluation? Could you ensure that it was always the most apical bone of the socket that was harvested? What about photos of samples with 5-7 months of stabilization? It would have been nice for comparison.

Thank you for your questions. The extraction was carried out on wisdom teeth whose roots no longer had contact with the alveolar canal. Each specimen was taken exactly from the newly formed bone, with the same trephine bur in each case (Lines 113-114), and its most apical portion was immediately stained on to assure a correct tissue orientation (Lines 117-121).

In order to collect the newly formed bone, the trephine bur orientation was guided by the main axis of the tooth movement, induced by the extrusive force. The location and axis planning for the biopsy were defined through the final CT scan. (Lines 118-119).

Do to the retrospective characteristic of this study, it is not possible anymore to achieve images of the 5-7 months period of stabilization. We are extremally sorry. However, we believe that the data reported are equally indicative of how the bone remodeled itself in that period of time.

- If this is a pilot study, what do the authors plan to do further?

Thank you for the question about our future intentions. Certainly, we want to continue this kind of analysis but first of all trying to implement a prospective protocol, so as to standardize the orthodontic procedure and plan the stabilization times in the best possible way.

Reviewer 2 Report

This study investigated bone modelling of neo-apposed tissue during orthodontic extrusive movements, through an histomorphometric analysis of human biopsies. The manuscript is well written, and the topic is interesting. However, there are some major issue which should be addressed. The current form is not acceptable:

1. Please add a background of previous studies in the introduction part.

2. The references and the literature discussed in the present study are so old. There are several studies on this subject since 2018 (the newest reference in your study). It is a major issue.

3. Kindly add the limitation of your study in the discussion part.

Author Response

This study investigated bone modelling of neo-apposed tissue during orthodontic extrusive movements, through an histomorphometric analysis of human biopsies. The manuscript is well written, and the topic is interesting. However, there are some major issue which should be addressed. The current form is not acceptable:

  1. Please add a background of previous studies in the introduction part.

Thank you for your comments, useful to improve the comprehension of the manuscript and to make it more updated. As requested, we have added the explanation of background studies related to OE in the Introduction part (Lines 64-71).

  1. The references and the literature discussed in the present study are so old. There are several studies on this subject since 2018 (the newest reference in your study). It is a major issue.

Thank you for your suggestion. We added more recent references (# 10, 26, 32 and 33), and we replaced the oldest ones (#8, 9 and 22).

  1. Kindly add the limitation of your study in the discussion part.

Thank you for your suggestion. Effectively it was an elusive part. This study presents several limits and, among them, we recognize the followings as the most relevant: sample size, retrospective design, histological analysis and heterogeneity of the extrusive technique. All these aspects are now presented into the discussion section (Lines 192-197 and 237-239).

Round 2

Reviewer 2 Report

I would like to thank the authors for discussing and considering my comments. There is only minor suggestion left:

The updated references are not still enough. Over 50 % of the references are older than 10 years

Author Response

Dear reviewer, thank you for giving us the chance to update our references.

After this second review, 22 out of 36 references are referred to recent articles published in the last 10 years. In detail, references #1-5, 7-11, 14, 21, 22, 26, 28-33 and 35,36 have not been published earlier than 2012.

We hope that this important references update will be considered sufficient to consider the new reference list recent and up-to-date.

Best regards